# Sensory and Biological Activity of Medlar (*Mespilus germanica*) and Quince 'Nivalis' (*Chaenomeles speciosa*): A Comperative Study

Anna K. Żołnierczyk [1,*], Natalia Pachura [1], Przemysław Bąbelewski [2] and Ebrahim Taghinezhad [3]

1 Department of Food Chemistry and Biocatalysis, Faculty of Biotechnology and Food Science, Wrocław University of Environmental and Life Sciences, Norwida 25, 50-375 Wrocław, Poland; natalia.pachura@upwr.edu.pl

2 Department of Horticulture, Wrocław University of Environmental and Life Sciences, Grunwaldzki 24A, 50-375 Wrocław, Poland; przemyslaw.babelewski@upwr.edu.pl

3 Department of Agricultural Technology Engineering, Moghan College of Agriculture and Natural Resources, University of Mohaghegh Ardabili, Ardabil 56199-11367, Iran

* Correspondence: anna.zolnierczyk@upwr.edu.pl

**Abstract:** This research investigates the potential health benefits of extracts from the seeds, peels, and pulps of quince, medlar, and bletting medlar fruits. Our study reveals that the polyphenol content is higher in the skin than in the flesh of the fruits tested, with the highest concentration found in the skin of fresh medlar fruits (1148 mg GAE/100 gDM). The extracts from medlar and quince show the highest antioxidant activity (ABTS, DPPH, and FRAP tests), while the pulp of bletting medlars exhibits the highest inhibition ability against α-amylase (53.7% at a concentration of 10 mg/mL). The analysis of fatty acids in the tested samples indicates the presence of nine major fatty acids, with linoleic acid being the most abundant (716–1878 mg/100 g of biomass). Analysis of sterols in the tested material shows five main phytosterols, with β-sitosterol being the most commonly studied and recommended phytosterol. The highest amount of phytosterols is found in the lipid fraction of the quince seeds (1337.1 mg/100 g of biomass). Therefore, we suggest that fruit peel extracts can be utilised as a natural source of antioxidants and as an alternative treatment for carbohydrate uptake disorders. However, it is important to note that bletting medlar loses a significant amount of polyphenols and antioxidant activity after the bletting process. This article also describes the sensory analysis process, which is a valuable tool for evaluating the quality of food products. Our study evaluates the attributes and preferences of the fruits of quince, medlar, and bletting medlar using a nine-point hedonic scale. The results show that quince is the highest-rated fruit in terms of aroma, colour, and overall acceptability (7.3, 7.0, and 4.2, respectively) while bletting medlar is the least preferred fruit. The article concludes that sensory analysis can aid in the development of new products and recipes that meet consumer preferences. In general, the study suggests that both fruit peel extracts and sensory analysis are important tools for assessing product quality and developing products that meet consumers' preferences.

**Keywords:** medlar fruit; quince fruit; *Mespilus germanica*; *Chaenomeles speciose*; antidiabetic activity; antioxidant activity; sensory analysis





## 1. Introduction

The food and pharmaceutical industries are dynamically developing sectors, constantly searching for new sources of health-promoting and biologically active substances in the world of fruits and vegetables. Medlar (*Mespilus germanica*) and quince (*Chaenomeles speciosa*) are both members of the Rosaceae family and produce edible fruits that are used in various culinary applications. Both are appreciated for their unique flavor and nutritional value. In terms of nutritional composition, medlar fruit is rich in vitamins C, A, and E, as

well as minerals such as iron and potassium, while quince fruit is a good source of vitamin C, fibre, and antioxidants [1–4]. One unique aspect of medlar fruits is that they are not typically eaten fresh. Instead, the fruit is usually left on the tree until it has softened and developed a slightly wrinkled appearance. This process, known as "bletting," allows the starch of the fruit to be converted into sugars, resulting in a sweeter and more flavorful fruit. Several studies have investigated the biological activity of medlar and quince extracts. For example, a study found that medlar extracts have antioxidant, antimicrobial, and antidiabetic effects in in vitro models [5,6]. Another study reported that quince extracts have high antioxidant and $\alpha$-glucosidase inhibitory activities [7]. Moreover, the essential oils of the quince fruit have been shown to have antimicrobial activity against various microorganisms [8]. In terms of their chemical composition, both medlar and quince contain a variety of bioactive compounds, including phenolic acids, flavonoids, and tannins [9–11]. These compounds are responsible for the antioxidant, anti-inflammatory, and antimicrobial activities observed in these fruits. *Chaenomeles speciosa* has the potential to serve as a strong source of anti-inflammatory and antiviral compounds. One such compound is quercetin, a potent antioxidant that may have the ability to serve as a candidate for anti-flu drugs [12]. However, the specific types and amounts of bioactive compounds may differ between the two fruits. An important group of compounds present in the fruits studied is phytosterols, which are plant compounds structurally similar to cholesterol. Incorporating them into the diet can have positive effects on cardiovascular health. Phytosterols can be found in a variety of foods, including nuts, seeds, grape seed oil, wheat germ oil, and vegetables such as broccoli and spinach [13]. Many food companies are adding phytosterols to products such as margarine, fruit juices, milk, and yoghurt to help lower blood cholesterol levels. Low-cost alternative sources of these valuable compounds are being sought, using by-products and waste from the agri-food industry, such as fruit pomace [14]. The fatty acids present in the oil fraction can exhibit antimicrobial activity against many bacteria, fungi, and viruses [15]. Unsaturated fatty acids, such as omega-3 and omega-6 fatty acids, are important components of the human diet. Their regular consumption can have a positive effect on heart health by reducing the risk of cardiovascular disease. They can also help lower blood triglyceride levels, improve brain function, have a positive effect on skin and hair health, and exhibit anticancer properties [16].

In general, both medlar and quince fruits are nutritious and have potential health benefits because of their bioactive compounds. Further studies are needed to better understand their biological activities and mechanisms of action, as well as to compare their nutritional and bioactive profiles.

The objective of this study was to investigate the biological activity of medlar fruits (*Mespilus germanica* L.) and quince 'Nivalis' (Chaenomeles speciosa 'Nivalis') fruits by analysing their fatty acid and sterol profiles, total polyphenol content, anti-diabetic, and antioxidant activities of their flesh, skin, and seed extracts. The effects of storage for one month on the aforementioned parameters of medlar fruits were also evaluated. The fruits studied were chosen for their rich source of bioactive compounds, and the biological tests allowed an evaluation of their potential health benefits. Sensory analysis provided information on consumer preferences and the potential to use the fruits studied in the food industry.

## 2. Materials and Methods

### 2.1. Plant Materials

The fruits of medlar (M) and quince (Q) were from a private farm in Down Silesia (Poland). The fruit was harvested in the second half of September 2021. The fruits were divided into three fractions: flesh (F), skin of the fruit (FS), and seeds (S); then they were freeze-dried. A portion of the medlar fruits was stored at a temperature of 15 °C for 30 days using a process known as 'bletting.' Subsequently, the fruits were divided into three fractions prior to being freeze-dried, similar to the fresh fruits: flesh (FB), fruit skin (FSB), and seeds (SB). Sample codes are listed in Table 1.

**Table 1.** Sample codes.

| No | Sample | Sample Code |
|----|--------|-------------|
| 1 | Medlar fruit flesh | MF |
| 2 | Medlar fruit skin | MFS |
| 3 | Medlar fruit seeds | MS |
| 4 | Bletting medlar fruit flesh | MFB |
| 5 | Bletting medlar fruit skin | MFSB |
| 6 | Bletting medlar fruit seeds | MSB |
| 7 | Quince fruit flesh | QF |
| 8 | Quince fruit skin | QFS |
| 9 | Quince fruit seeds | QS |

*2.2. Extraction Procedure*

The fruits of medlar, bletting medlar, and quince (2.0 kg) were divided into flesh, fruit skin, and seeds and then lyophilized and ground in an electric mill. The obtained material was extracted with an 80% aqueous ethanol solution acidified with hydrochloric acid (10 mL HCl per 1 L of solution) or Folch solvent. Of the solvent solution, 10 mL per gramme of dry mass was used. The extraction was carried out on a rotary shaker for 24 h at 60 °C (for ethanolic extracts) and for 24 h at 25 °C (for Folch extracts). The organic solvent was evaporated on a rotary evaporator, and the residue was freeze-dried. Figure 1 shows the masses of research material (quince, medlar, and bletting medlar fruits) and obtained extracts.

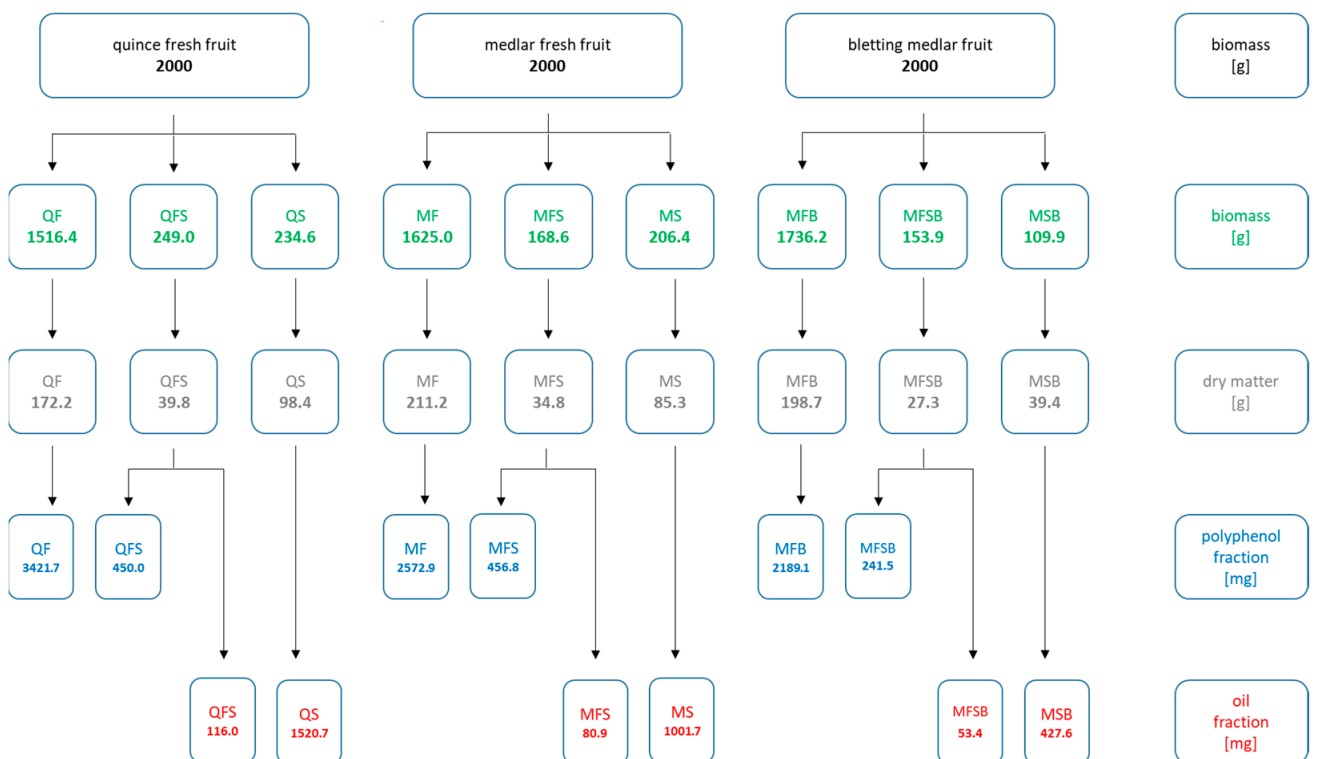

**Figure 1.** Masses of the research material and obtained extracts.

*2.3. Analysis of Antioxidant Activity and Total Phenol Content*

The evaluation of antioxidant activity was carried out using modified procedures for the methods of ABTS [17], DPPH [18], and FRAP [19]. The absorbance readings were taken at specific wavelengths, namely 734 nm for ABTS, 517 nm for DPPH, and 563 nm for the FRAP method, using a UV Microplate Reader μQuant (Bio Tek, Winooski, VT, USA). The entire investigation was conducted in triplicate. The amount of antioxidant

activity was reported as μM of Trolox equivalent per g of dry extract. Total polyphenols were determined using the modified Folin–Ciocalteu method [20]. All determinations were performed in triplicate. The results were expressed as milligrammes of gallic acid equivalent (GAE) per 100 g of dry matter (DM).

### 2.4. Determination of Antidiabetic Activity

To evaluate antidiabetic activity, the diffusion method was used [21,22]. Samples of the obtained materials, namely extracts of MF, MFS, MS, MFB, MFSB, MSB, QF, QFS, or QS, were dissolved in 1 mL of dimethyl sulfoxide (DMSO) at concentrations of 10, 25, or 50 mg. The resulting powder was completely dissolved, and cylindrical wells of 0.5 cm diameter were cut into the agar medium (3%) containing 1% starch, which had been previously prepared, poured into Petri plates, and stored at 4 °C. The negative control (T−) consisted of a solution of 25 μL of acarbose solution (50 mg/1 mL $H_2O$)—a compound found in antidiabetic drugs—and 25 μL of pork α-amylase solution (6 mg/10 mL $H_2O$). The positive control (T+) was a solution of 25 μL of water and 25 μL of pork α-amylase solution. The tests involved the introduction of 25 μL of the tested medlar and quince extracts and 25 μL of pork α-amylase solution into the agar wells. After 24 h of incubation at 37 °C, the plates were treated with iodine vapours, and the resulting clear zones were measured to calculate the degree of inhibition. The clear zone produced by the positive control (T+) was considered to represent 100% inhibition. The experiments were carried out in triplicate.

### 2.5. Determination of Fatty Acid Composition

The freeze-dried plant material was extracted using a chloroform–methanol mixture (2:1; *v:v*). The resulting extracts were evaporated under a vacuum and then subjected to basic hydrolysis and methylation using KOH/MeOH/BF$_3$ [23]. The prepared samples were dissolved in 2 mL of hexane, and the fatty acid composition was analysed using a gas chromatograph with mass detection (GC/MS). The compounds were identified by comparing the obtained spectra with the NIST14 database and the retention times of the standards (Supelco 37 Component FAME Mix, certified reference material, TraceCERT®, in dichloromethane (varied conc.)). The GC–MS analysis was carried out using a ZB-WAXplus column (30 m × 0.25 mm × 0.25 μm) on a Shimadzu GCMS-QP2020. The split was set at 100:1, and helium was used as the carrier gas at a flow rate of 1.0 mL/min in constant flow mode. The injector temperature was 260 °C, and the column ramp temperature was 160 °C (for 5 min), then increased to 200 °C (at a rate of 2 °C/min) and to 250 °C (at a rate of 10 °C/min), and held for 4 min. The entire analysis took 34 min. A full scan was performed in the range of 50–500 AU with 0.5 scan s$^{-1}$. The quantification of compounds was based on the area of the signal, without the application response factor.

### 2.6. Determination of the Sterol Profile

The samples extracted and evaporated according to Section 2.2 underwent BSTFA derivatization [24]. They were then dissolved in 2 mL of hexane and subjected to gas chromatography–mass spectrometry (GC/MS) analysis to determine the sterol profile. Phytosterol compounds were identified by comparing their spectra with those of the NIST14 database and using the retention times of available standards (Aldrich). The GC–MS analysis was performed on a Shimadzu GCMS-QP2020 using a ZB-5 column (30 m × 0.25 mm × 0.25 μm, from Zebron, Phenomenex, Shim-Pol,Warsaw, Poland). The split ratio was set at 10:1, and helium was used as the carrier gas with a constant flow rate of 1.0 mL/min. The injector temperature was set at 280 °C, and the column temperature was ramped from 170 °C to 300 °C (a buildup of 5 °C/min) with an analysis end at 300 °C. The entire analysis took 36 min, and the detector temperature was set at 250 °C, with a full scan in the range of 40 to 500.

*2.7. Sensory and Aroma Analysis*

A nine-hedonic scale (Table 2) was used to investigate the degree of preference for the quince, common medlar, and bletting medlar fruits. The three treatments were evaluated in one session in a specially designed laboratory. Ten panelists were chosen from the teaching staff, graduate students, and master's degree students of the Faculty of Biotechnology and Food Science, Wrocław University of Environmental and Life Sciences (nine trained assessors). The panelists were of both sexes and different ages. The fruit samples were served on small plates. After each sample, the panelists drank water to restore their original tasting conditions. The odour descriptors were determined in preliminary tests. A set of reference solutions in water (0.01–0.1%; concentrations well above the threshold but evaluated as not very intense) was prepared based on the set of odour descriptors, consisting of caramel (no. 01), honey (no. 02), pear (no. 117), cinnamon (no. 18), vanilla (no. 30), and mousse (no. 97) (Sosa Ingredients, S. L., Barcelona, Spain). The descriptors for the evaluation of the fruits tested were designated from the data in the literature. The panelists were trained for three one-hour sessions. To assist panelists in establishing a framework for each attribute, reference smells were used during training to establish minimum and maximum intensities for each attribute. The sensory quality of the samples was evaluated in sweet, floral, honey, spicy, and grassy categories using a varying scale from 9—which means extreme like to 1—which means extreme dislike (Table 2).

**Table 2.** Nine-point hedonic scale is used in the preference test.

| Grade | Score |
|---|---|
| Like extremely | 9 |
| Like very much | 8 |
| Like moderately | 7 |
| Like slightly | 6 |
| Neither like or dislike | 5 |
| Dislike slightly | 4 |
| Dislike moderately | 3 |
| Dislike very much | 2 |
| Dislike extremely | 1 |

Tested fruits were subjected to sensory evaluation by an untrained panel formed of 40 panelists between 19 and 26 years of age to test for the acceptability and preferences of potential consumers. All leathers were tested at the same time. The sensory attributes evaluated were tartness, sweetness, aroma, firmness, colour, and overall acceptance using a nine-point hedonic scale (Table 2).

*2.8. Statistical Analysis*

Data were analysed using Statistica 13 software (Kraków, Poland). Tukey's (HSD) test analysed the differences between the means (*p*-value < 0.05). The tables present the average standard deviations.

**3. Results and Discussion**

Figure 2 shows a block diagram of the experiments carried out. Figure 1 presents the masses of research material (quince, medlar, and bletting medlar fruits) and obtained extracts. The total amount of polyphenols was expressed as milligrammes of gallic acid equivalent per 100 grammes of the dry weight of the tested material (Figure 3). Higher concentrations of these compounds were found in the skin of the tested fruits than in the flesh—with quince showing a 15% increase and bletting medlar showing a 28% increase, which is consistent with previous scientific literature [25]. The highest content of polyphenols was found in the skin of fresh medlar fruits.

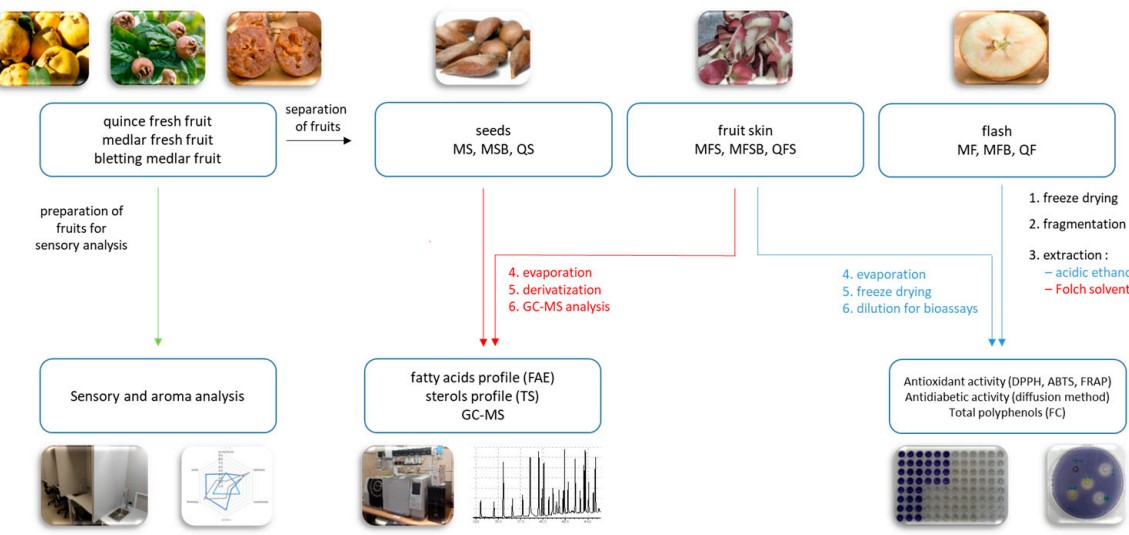

**Figure 2.** Block diagram of conducted experiments.

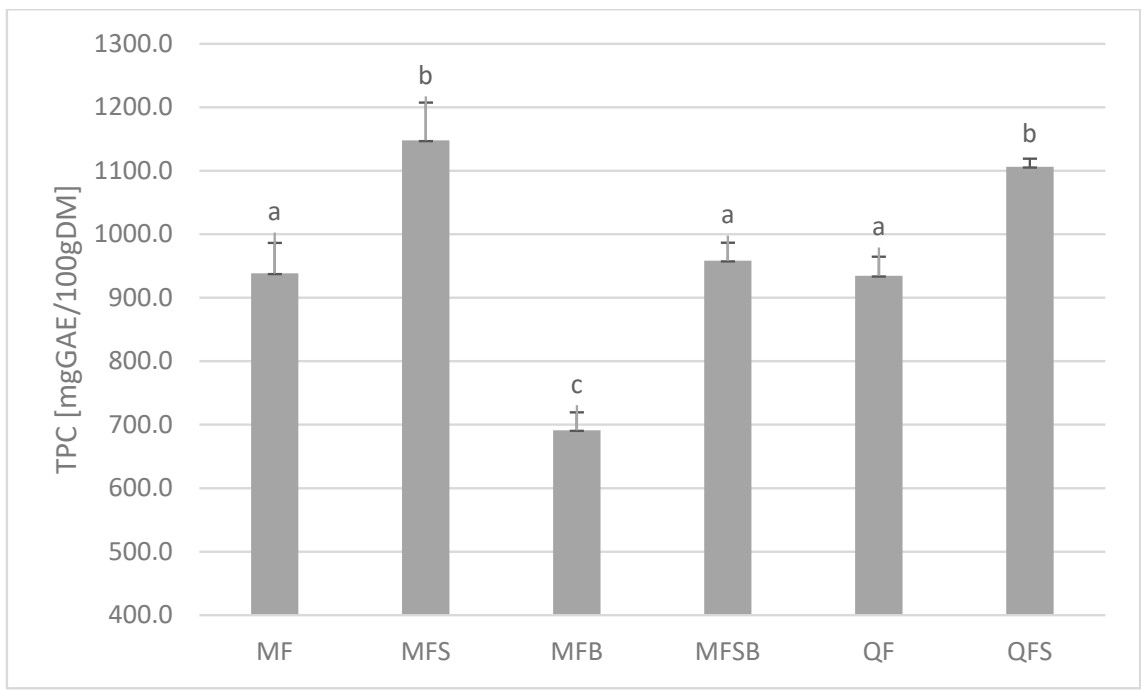

**Figure 3.** Total phenol content (TPC) (mg GAE/100 gDM). Mean values with different letters. (a–c) within the same column were statistically different ($p < 0.05$), and the same letters form one homogeneous group.

Fruit peel is often discarded and not consumed, despite being a rich source of polyphenols, which are plant-derived compounds with potential health benefits. Polyphenols have been shown to have antioxidant, anti-inflammatory, anti-cancer, and neuroprotective effects. However, most studies on polyphenols in fruits have focused on the flesh or pulp, with little attention paid to the peel. Several studies have reported high concentrations of polyphenols in fruit peel. For example, pomegranate fruit peel has been found to contain up to 21,690 mg GAE/100 gDM [26]. An important result of the research conducted is the significant decrease in the amount of polyphenols in the pulp and skin of bletting medlars compared to fresh fruits (by 26% in the flesh (MF, MFB) and 16% in the skin (MFS, MFSB), Figure 3). Unfortunately, bletting medlar is more acceptable for consumption in terms

of hardness (they soften over time) and texture (they become creamier), but this process results in the loss of valuable nutritional sources.

The DPPH, ABTS, and FRAP methods are widely used for studying the antioxidant activity of plant extracts, enabling the determination of the ability of the extracts to neutralise free radicals, which is associated with their antioxidant properties [27,28]. The DPPH method is based on the molecule's ability to reduce the free 2,2-diphenyl-1-picrylhydrazyl radical; the ABTS method measures the substance's ability to reduce the free 2,2′-azino-bis(3-ethylbenzothiazoline-6-sulfonic acid radical; and the FRAP method is based on the substance's ability to reduce $Fe^{3+}$ ions to $Fe^{2+}$.

Table 3 shows the antioxidant activity of the tested dry plant extracts measured by DPPH, ABTS, and FRAP methods, expressed as Trolox equivalent antioxidant capacity (μm). Trolox is used as a standard reference antioxidant compound. The results are presented as mean values with standard deviation. The highest DPPH activity was observed in the MFS extract (510.7 ± 59.5 μm), followed by MF (453.3 μm TEAC) and QFS (412.3 μm). The highest ABTS activity was found in QFS (1058.0), followed by MFS (943.0), and MF (T832.0). The highest FRAP activity was observed in MFS (887.0 ± 43.3 μm), followed by MF (829.7 μm) and MFSB (659.7 μm). Among the plant extracts tested, MFS and QFS exhibited the highest antioxidant activity as measured by the DPPH, ABTS, and FRAP methods. These findings suggest that these extracts could be used as potential sources of natural antioxidants in the food and pharmaceutical industries. Similarly, in the determination of the polyphenol content, the antioxidant activity decreases after the medlar bletting process. In general, these results suggest that MFS and MF extracts possess high antioxidant activity according to the DPPH and ABTS methods, while MFS and MFSB extracts have high FRAP activity. The QFS extract was found to have the highest ABTS activity among all the extracts tested.

**Table 3.** The antioxidant activity: Trolox equivalent antioxidant capacity is defined as the concentration of Trolox (μm) with the same activity as 1 g of the tested dry extract (the DPPH, ABTS, and FRAP methods).

| No. | Plant Material | DPPH | ABTS | FRAP |
| --- | --- | --- | --- | --- |
| 1 | MF | 453.3 ± 21.8 b,c | 832.0 ± 31.0 a,c | 829.7 ± 35.9 b |
| 2 | MFS | 510.7 ± 59.5 c | 943.0 ± 39.0 c | 887.0 ± 43.3 b |
| 3 | MFB | 309.0 ± 48.5 a | 719.0 ± 20.4 b | 452.0 ± 48.2 a |
| 4 | MFSB | 323.3 ± 61.3 a,b | 735.3 ± 52.8 a,b | 659.7 ± 36.8 c |
| 5 | QF | 298.3 ± 36.2 a,b | 819.7 ± 23.5 a | 402.7 ± 30.4 a |
| 6 | QFS | 412.3 ± 23.3 a,c | 1058.0 ± 29.2 d | 445.7 ± 45.3 a |

Mean values for different letters (a–d) within the same column were statistically different ($p < 0.05$), the same letters form one homogeneous group. Values are expressed as the mean ± standard deviation.

The results in terms of antidiabetic activity (Table 4) indicate that extracts of the pulp and skin of quince, medlar, and bletting medlar fruits at a concentration of 100 mg per mL inhibit the activity of α-amylase by 100%. The extract of bletting medlar pulp (MFB) showed the highest activity against the enzyme, with an ability to inhibit over 50% at a concentration of 10 mg/mL. A strategy to treat carbohydrate uptake disorders, such as diabetes and obesity, involves inhibiting α-amylase—an enzyme that aids in the digestion of starch and glycogen [29]. Its inhibitors contribute to the reduction of postprandial hyperglycemia and the slowing of carbohydrate digestion in people with diabetes. Numerous scientific studies have shown that plant extracts with health-promoting effects in the human body also exhibit antidiabetic activity by inhibiting the activity of α-amylase. Chemical compounds derived from plants have the potential to inhibit α-amylase and can be used as therapeutic agents or functional food sources. The compounds responsible for this activity are usually flavonoids [30,31] but extracts rich in triacylglycerols also show potential [32,33].

**Table 4.** Antidiabetic activity of the tested extracts expressed in % inhibition of $\alpha$-amylase.

| | | Percentage of Inhibition | | | |
|---|---|---|---|---|---|
| **No.** | **Plant Material** | Concentration [mg/mL DMSO] | | | |
| | | **100** | **50** | **25** | **10** |
| 1 | MF | 100 ± 0.0 a | 100 ± 0.0 b | 38.3 ± 2.5 b | 0.0 ± 0.0 a |
| 2 | MFS | 100 ± 0.0 a | 46.3 ± 3.9 c | 0.0 ± 0.0 a | 0.0 ± 0.0 a |
| 3 | MFB | 100 ± 0.0 a | 100 ± 0.0 b | 100 ± 0.0 e | 53.7 ± 1.2 b |
| 4 | MFSB | 100 ± 0.0 a | 63.9 ± 3.0 a | 43.0 ± 1.9 c | 0.0 ± 0.0 a |
| 5 | QF | 100 ± 0.0 a | 89.3 ± 3.5 d | 55.4 ± 1.2 d | 0.0 ± 0.0 a |
| 6 | QFS | 100 ± 0.0 a | 56.6 ± 2.3 a | 0.0 ± 0.0 a | 0.0 ± 0.0 a |

Mean values for different letters (a–e) within the same column were statistically different ($p < 0.05$); the same letters form one homogeneous group. Values are expressed as the mean ± standard deviation.

Nine major fatty acids (FAs) were identified in the skin samples of fruits and seed oils using the GC/MS method. The fatty acid profile is presented in Table 5. In the observed analyses, lower amounts of fatty acids were observed in the fruit skins (MFS, MFSB, and QFS) than in the seeds (MS, MSB, and SQ). Palmitic, stearic, and linoleic acids were present in all analyses, and linoleic acid was the most abundant. The MS and QS samples had the highest content of saturated fatty acids (SFA), with 1692 and 1517 mg per 100 g of biomass, respectively. QFS had the highest amount of monounsaturated fatty acids (MUFA), at 1990 mg/100 g, while MS had the highest content of polyunsaturated fatty acids (PUFA), at 2694 mg/100 g. The skin and seeds of the medlar (MFS, MS) did not contain MUFA. Previous literature reports confirm that the fat fraction obtained from residues of Japanese quince (*Chaenomeles japonica*) and medlar fruit (*Mespilus germanica* 'Dutch') is rich in PUFA, mainly ω-6 linoleic acid [34,35]. In addition, important nutrients (carbohydrates, organic acids, and fatty acids) of the medlar fruit are lost during the bletting process, which is nutritionally significant [35].

**Table 5.** Fatty acid profile quantified (mg/100 g) in the biomass of quince and medlar fruit skin and seeds.

| No. | Fatty Acid * | | MFS | MS | MFSB | MSB | QFS | QS |
|---|---|---|---|---|---|---|---|---|
| 1 | C 12:0 | Lauric acid | n.d. | n.d. | n.d. | 23.7 ± 4.2 a | n.d. | n.d. |
| 2 | C 14:0 | Mirstic acid | n.d. | n.d. | n.d. | 55.0 ± 5.6 a | n.d. | n.d. |
| 3 | C 16:0 | Palmitic acid | 347.7 ± 24.4 a | 938.7 ± 25.1 c | 425.7 ± 32.1 a | 604.7 ± 9.1 b | 598.0 ± 53.9 b | 1341.3 ± 70.5 d |
| 4 | C 18:0 | Stearic acid | 443.7 ± 36.8 b | 753.7 ± 43.9 c | 152.7 ± 18.0 a | 99.3 ± 10.0 a | 74.0 ± 7.6 a | 105.47 ± 51.2 a |
| 5 | C 18:1 | Elaidic acid | n.d. | n.d. | 20.3 ± 4.5 a | 33.0 ± 4.6 a | 78.0 ± 10.1 a | n.d. |
| 6 | C 18:1 | Oleic acid | n.d. | n.d. | 622.7 ± 25.3 a | 900.3 ± 38.8 b | 1912.0 ± 80.1 c | 1754.7 ± 51.2 b |
| 7 | C 18:2 | Linoleic acid | 1855.7 ± 52.2 a | 1878.0 ± 42.5 a | 716.0 ± 65.6 b | 878.0 ± 40.0 c | 1869.0 ± 48.1 a | 1942.7 ± 22.0 a |
| 8 | C 18:3 | α-Linolenic acid | 142.7 ± 10.7 a | 816.3 ± 18.2 b | 71.3 ± 11.2 a | 363.7 ± 37.5 c | n.d. | 867.3 ± 59.7 b |
| 9 | C 20:0 | Arachidic acid | n.d. | n.d. | n.d. | n.d. | n.d. | 70.0 ± 8.6 a |
| 10 | | Total | 2789.8 | 4386.7 | 2008.7 | 2957.7 | 4531.0 | 6081.2 |

* Expressed as methyl esters according to the GC–MS chromatogram. a–d: homogeneous groups according to Tukey's test. Values are expressed as the mean ± standard deviation. n.d. = not determined.

From the oils analysed, the linoleic acid present in the highest amount is a polyunsaturated fatty acid from the omega-6 family, also known as CLA, and must be obtained through the diet. Studies indicate that it may help lower LDL cholesterol levels and increase HDL cholesterol levels [36,37], which may contribute to improved cardiovascular function. The role of CLA in cancer prevention is well established, as it effectively inhibits all stages of carcinogenesis: initiation, promotion, and metastasis [38]. It has been demonstrated that dietary supplementation with selected isomers of CLA results in a reduction of adipose tissue. This reduction has been shown in various animal [39,40] and human [41,42] models. Furthermore, CLA has been found to exhibit anti-inflammatory [43] and antidiabetic [44] properties. Studies have shown that the consumption of CLA may be beneficial for health, but doses and effects may vary depending on the individual and the purpose of consumption. It is important to note that there are no clear guidelines for CLA doses, and the effects

of its consumption may vary depending on individual needs and health conditions [45,46]. Before starting CLA supplementation, it is always advisable to consult with a doctor or nutrition specialist to determine the appropriate dose and prevent potential side effects.

In the analysed material, an analysis of sterols in the fat fraction was performed. On the basis of the GC-MS results, the presence of five major phytosterols was found (Table 6). The highest amount of phytosterols in terms of quality (four out of five) and quantity (1337.1 mg/100 g) was identified in the QS lipid fraction. The highest percentage in this fraction (similar to the other extracts) was β-sitosterol—more than 80%. β-sitosterol is the most commonly studied and recommended phytosterol. During the bletting process, the content of phytosterols decreased by 36% in the skin of the fruit (MFSB) and by 56% in the seeds (MSB). Scientific studies show that the consumption of beta-sitosterol can lower LDL levels in the blood and reduce the risk of cardiovascular disease [47]. For example, beta-sitosterol supplementation helped reduce LDL cholesterol levels in individuals with hypercholesterolemia. Other studies suggest that consuming 2–3 g of beta-sitosterol per day may have a beneficial effect on blood cholesterol levels in people with high cholesterol levels [48]. However, excessive consumption of phytosterols can cause disruptions in the absorption of certain nutrients, so it is important to maintain moderation in the diet.

**Table 6.** Sterol profile (mg/100 g) of the biomass of quince and medlar fruit skin and seeds.

| No. | Sterols | MFS | MS | MFSB | MSB | QFS | QS |
|-----|---------|-----|-----|------|-----|-----|-----|
| 1 | Campesterol | n.d. | n.d. | n.d. | 53.0 ± 12.5 a | 219.7 ± 24.5 b | n.d. |
| 2 | unknown | n.d. | n.d. | n.d. | n.d. | 37.7 ± 9.1 a | 136.0 ± 8.2 b |
| 3 | Stigmasterol | n.d. | n.d. | n.d. | 36.3 ± 11.7 a | n.d. | 42.7 ± 11.1 a |
| 4 | β-Sitosterol | 500.7 ± 47.6 a | 927.0 ± 62.1 c | 319.0 ± 21.3 b | 406.0 ± 16.1 a,b | 461.7 ± ±39.1 a | 1095.7 ± 50.5 d |
| 5 | Cycloartenol | 70.3 ± 12.7 a | n.d. | 74.0 ± 16.1 a | n.d. | n.d. | 62.7 ± 12.0 a |
| 6 | Total | 571.0 | 927.0 | 393.0 | 495.3 | 719.1 | 1337.1 |

Mean values for different letters (a–d) within the same row were statistically different ($p < 0.05$); the same letters form one homogeneous group. Values are expressed as the mean ± standard deviation. n.d. = not determined.

Phytosterols are plant compounds that are structurally similar to cholesterol. When consumed in the diet, they can limit the absorption of cholesterol from the intestines and thus affect the level of cholesterol in the blood [49]. According to scientific reports, phytosterols may help reduce the risk of cardiovascular disease by lowering the level of LDL in the blood [50]. Studies suggest that phytosterol supplementation can lower LDL levels by approximately 10–20%, but its effectiveness can vary depending on the dose, duration of supplementation, and other factors [51]. Phytosterols are also used in the food industry as a component of cholesterol-lowering food products. However, excessive consumption of phytosterols can cause disturbances in the absorption of certain vitamins (such as vitamin E) and other nutrients [52]. It is important to pay attention to a balanced diet, of which phytosterols are only one component. The introduction of plant-sterol-enriched food products, which are increasingly popular among consumers, can contribute to reducing the risk of cardiovascular disease. Plant-based sterol-enriched margarine, mayonnaises, oils, sauces, and other food products that are readily available to everyone can provide a healthier alternative to high-calorie food products that are consumed in large quantities [53,54].

Sensory analysis is a scientific discipline that focuses on evaluating the quality of food products. This approach enables us to describe various attributes of food, including its visual appearance, scent, texture, and taste. The primary tools used in this analytical process are the human senses, which serve as measuring instruments for assessing food characteristics. In other words, sensory analysis is a branch of science that specialises in evaluating the excellence of food. By using our senses of sight, smell, touch, and taste, we can define various aspects of food, such as its visual appeal, aroma, mouthfeel, and overall taste. These senses act as "equipment" to measure and assess food properties. The sensory panel evaluated quince, medlar, and bletting medlar fruits using a nine-point hedonic scale, where one represented "extreme dislike" and nine represented "extreme like" (Table 2). The aim of the sensory analysis was to investigate the sensory attributes and overall preferences

of fresh fruits, including quince, medlar, and bletting medlar. According to the results of the sensory evaluation (Figure 4), quince was the fruit with the highest rating in terms of aroma, colour, and overall acceptability, receiving scores of 7.3, 7.1, and 7.1, respectively. Medlar was rated moderately acceptable in all categories, receiving scores of 4.2, 5.4, and 5.4 for aroma, colour, and overall acceptability, respectively. Bletting medlar was the least preferred fruit, receiving the lowest scores in all categories except for firmness, where it received the highest score of 7.7. Specifically, it received scores of 3.4 for tartness, 5.2 for sweetness, 3.1 for aroma, 2.9 for colour, and 2.9 for overall acceptability. Overall, the results of this study provide valuable information on the sensory attributes of these fruits and can aid in the development of new products and recipes that meet consumer preferences.

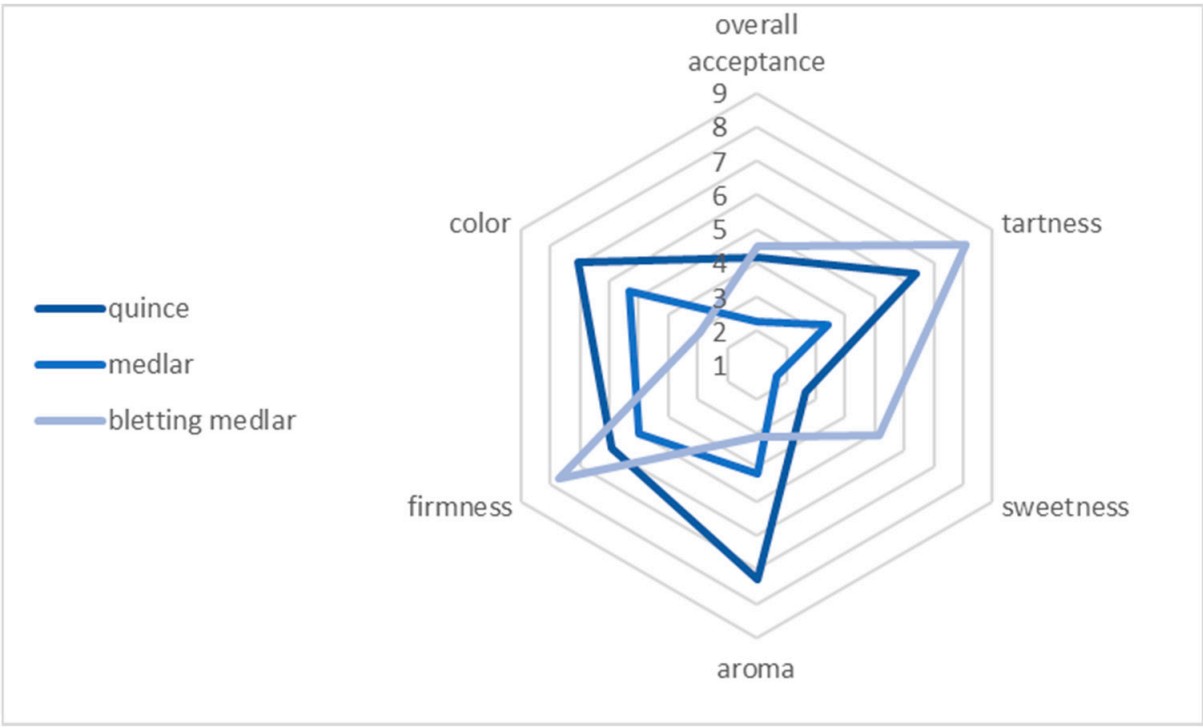

**Figure 4.** Sensory attributes of quince, medlar, and bletting medlar fruits.

The "spiderweb" graph in Figure 5 presents the average intensity ratings of the aroma attributes for the fruits tested. The attributes evaluated were sweet, floral, honey, spicy, and grassy. Significant differences were observed among the fruits tested in terms of floral, honey, and grassy aromas. Quince fruits were characterised by intense floral and honey aromas, achieving scores of 8.2 and 7.5, respectively. On the other hand, fresh medlar fruits were distinguished by their grassy aroma (7.1). Furthermore, the analysed fruits were not characterised by a spicy aroma (average scores of 4). During aroma analysis, quince fruits were rated the highest, while bletting medlar fruits were rated the lowest. Sensory aroma analysis is an important tool to assess product quality and can be applied in various fields, such as the food and cosmetics industries.

Common medlar and quince fruits are not usually suitable for raw consumption. This is confirmed by sensory data collected and literature reports [55]. Medlar fruits can be eaten raw but are typically consumed after freezing or storing in a dark place for a few days, which helps the ripening process and reduces their astringency. Medlar fruits are usually eaten after the flesh softens and turns into a jelly-like consistency. Before eating them raw, it is necessary to remove the skin and seeds, as they are difficult to digest and can be dangerous to humans. Quince fruits are typically very hard and sour in taste, so they are not commonly consumed raw. However, depending on the variety and ripeness of the fruits, quinces can be eaten raw, but they are best used for processing into jams, preserves, juices, or baked goods. To consume quince fruits raw, it is best to choose ripe fruits that are

soft and smell sweet. Quince fruits can be peeled, pitted, and sliced, but they will still be quite sour and hard. To soften their taste, one can try drizzling them with honey or sugar.

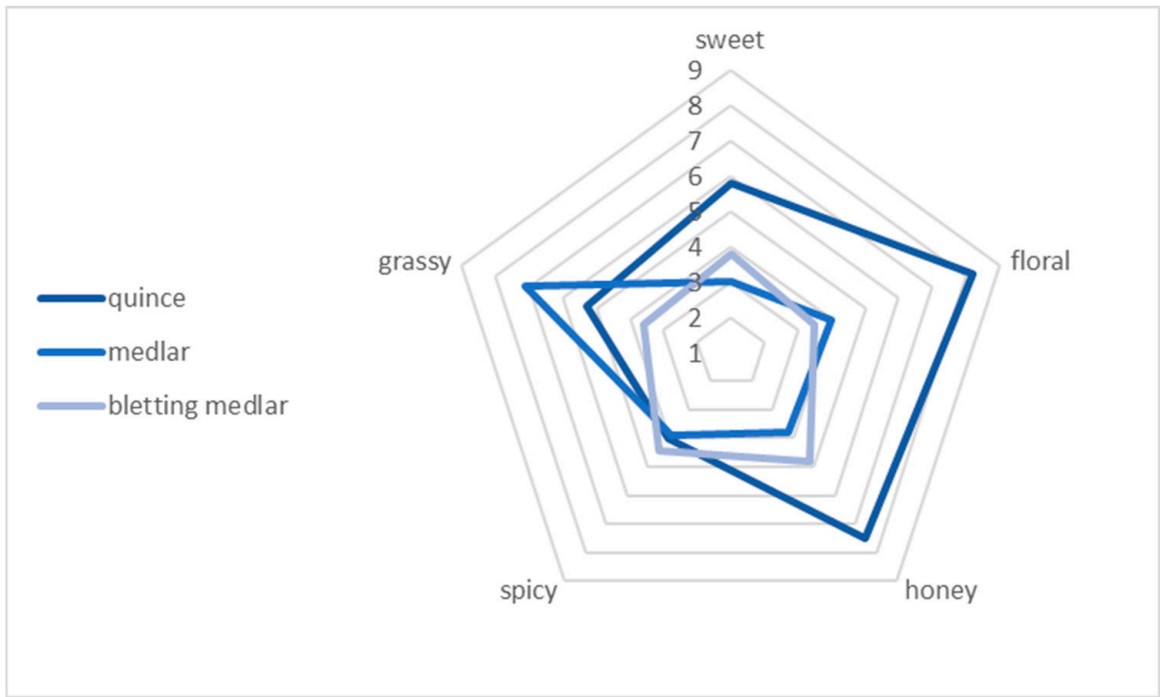

**Figure 5.** Aroma profile of the quince, medlar, and bletting medlar fruits.

## 4. Conclusions

The utilisation of agricultural by-products, such as fruit and vegetable waste, including seeds and pomace, appears to be a promising source of bioactive compounds with potential health benefits. Furthermore, the utilisation of the aforementioned waste materials aligns with the concept of a circular economy, which emphasizes reducing waste production and increasing resource efficiency [56,57].

The research conducted has shown that the seeds and skins of the studied materials stand out for their high content of linoleic acid. This information can be used to use seeds, which often constitute production waste, as an important source of oilseed. The aforementioned morphological parts of the fruits studied also showed a high content of β-sitosterol, thus creating the possibility of their use in the production of natural dietary supplements as well as nutritional components added to enrich foods. Furthermore, extracts obtained from the skins of the fruits studied exhibit antioxidant and antidiabetic properties, which gives them the potential to be used in the prevention and treatment of diseases.

Furthermore, the data presented in this work show that the compounds studied exhibited a tendency to decrease as the maturity of the medlar increased. Therefore, it is worth knowing the biochemical changes and activity of certain enzymes in more detail in order to carefully plan the harvesting and storage period while obtaining a sensory-attractive and nutritionally valuable raw material. With regard to the food and pharmaceutical industries, attention should be paid to the use of the medlar at the appropriate ripeness stage.

**Author Contributions:** A.K.Ż.—conceptualization, formal analysis, investigation, methodology, writing—original draft; review and editing; N.P.—software, statistical analysis; P.B.—plant material; E.T.—analysis. All authors have read and agreed to the published version of the manuscript.

**Funding:** This research received no external funding.

**Institutional Review Board Statement:** Not applicable.

**Data Availability Statement:** Data is contained within the article.

**Acknowledgments:** We acknowledge and would like to express gratitude to Milena Gajewska and Agata Jarosz for their valuable assistance in preparing the analyses.

**Conflicts of Interest:** The authors declare no conflict of interest.

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
