# Peer review of "Sensory and Biological Activity of Medlar (Mespilus germanica) and Quince ‘Nivalis’ (Chaenomeles speciosa): A Comperative Study"

_agriculture, doi:10.3390/agriculture13050922_

Round 1

Reviewer 1 Report

Dear authors,

The MS entitled (Sensory and biological activity of medlar (Mespilus germanica) and quince ‘Nivalis’ (Chaenomeles speciosa): a comperative study) has been reviewed

- The main question addressed by the research is the investigation of the potential health benefits of fruit peel extracts from quince, medlar, and bletting medlar fruits. 

The topic is original and relevant in the field.

- The results well presented and discussed

The conclusions consistent with the evidence and arguments the findings

and they address the main question.

- The references appropriate, mostly new and support the discussion of the findings.

However extensive English editing is required.

Also, authors have to justify the choice of plant material and the performed tests.

Author Response

Anna K. Żołnierczyk, PhD

Wrocław University of Environmental and Life Sciences

Department of Food Chemistry and Biocatalysis

Norwida 25, 50-375 Wrocław , Poland

anna.zolnierczyk@upwr.edu.pl

Thank you for your valuable comments regarding the manuscript entitled  Sensory and biological activity of medlar (Mespilus germanica) and quince ‘Nivalis’ (Chaenomeles speciosa): a comperative study. We appreciate your detailed review and hope that our statments will find your acceptance.

Dear Reviewer (1),

  • However extensive English editing is required.

We conducted a thorough linguistic analysis, and the changes were applied to the text.

  • Also, authors have to justify the choice of plant material and the performed tests.

The research on medlar and quince fruits was motivated by the fact that these fruits are rich in polyphenols and bioactive compounds that may contribute to beneficial effects on human health. The study focused on the polyphenol content in the fruit skins, as previous research has shown that the skins contain the highest amount of polyphenols.

Performing biological tests, such as antioxidant and anti-diabetic activity assays, allowed an evaluation of the potential health benefits of the fruits studied in the work. Antioxidant activity is important because it contributes to neutralizing free radicals, preventing cell damage and aging processes. Anti-diabetic tests allowed an assessment of the fruits influence of the studied on glucose metabolism in the body, which is significant in the context of the growing number of cases of diabetes.

Furthermore, conducting sensory analysis allowed for an evaluation of consumers' preferences regarding taste, aroma, and overall perception of the studied fruits. This provided knowledge about consumer preferences and the potential to use the studied fruits in the food industry to create products that match their taste.

In summary, the fruits studied were chosen because of their rich source of bioactive compounds and the conducted biological tests allowed for an evaluation of their potential health benefits. Sensory analysis provided information on consumer preferences and the potential to use the studied fruits in the food industry.

At the end of the Introduction chapter, a brief rationale for the selection of the fruits studied and the research conducted is included.

Reviewer 2 Report

Abstract

This study is on the health potential of fruit peels? Yet the indepth study was on the fruit pulp? 

Introduction 

How much oil is in the fruit pulp?

What is the purpose of this study? The background speaks of antioxidant capacity but not the sterols and fatty acids , their potential health benefit and if there is any link between these sterols and antioxidant activity? 

Methodology

The study goes on to determine total phenolic content, antioxidant activity and then sterols and fatty acids. What is the relationship between the sterols and antidiabetic activity?  

How much fat is in the skin and seeds? Why were these analysed? 

Results and discussion 

Is there any correlation between the linoleic acid and antidiabetic activity of the medler fruit?

Line 362-364 if the skin is dangerous when raw why are you analysing it raw? Wasn't it supposed to be processed first for analysis? Stored fruits are still raw. 

Conclusion 

Is this study just to profile the fatty acids and sterols in medlar because the antioxidant activity and antidiabetic activity are well known (line 52-60)?  Then the introduction should not be about antioxidant activity due to polyphenols but should justify the analysis of sterols and fatty acids and the conclusion should be on the purpose of this study.  

Author Response

Anna K. Żołnierczyk, PhD

Wrocław University of Environmental and Life Sciences

Department of Food Chemistry and Biocatalysis

Norwida 25, 50-375 Wrocław , Poland

anna.zolnierczyk@upwr.edu.pl

Thank you for your valuable comments regarding the manuscript entitled  Sensory and biological activity of medlar (Mespilus germanica) and quince ‘Nivalis’ (Chaenomeles speciosa): a comperative study. We appreciate your detailed review and hope that our statments will find your acceptance.

Dear Reviewer (2),

Abstract

  • This study is on the health potential of fruit peels? Yet the indepth study was on the fruit pulp?

The research carried out focused on the health potential of the oil fraction obtained from the peels and seeds of the fruits tested (medlar and quince), as the well as alcoholic extracts from the pulp and peels (which are rich in polyphenols). This has been corrected in the Abstract.

Introduction

  • How much oil is in the fruit pulp?

There is not much oil in the flesh of the fruits tested and its content is usually negligible. Most of the oil in medlar and quince fruits is found in the seeds and skin, although even there the amount is relatively low. Therefore, we decided to isolate the oil from the peel and seeds of the studied fruits.

  • What is the purpose of this study? The background speaks of antioxidant capacity but not the sterols and fatty acids , their potential health benefit and if there is any link between these sterols and antioxidant activity?

I supplemented the introduction with information on the biological properties of phytosterols and fatty acids.

Methodology

  • The study goes on to determine total phenolic content, antioxidant activity and then sterols and fatty acids. What is the relationship between the sterols and antidiabetic activity?

Numerous studies have proven that plant extracts rich in anthocyanins, which are responsible for pro-health effects in the human body, also show anti-diabetic activity by inhibiting α-amylase and α-glucosidase activity. However, in the case of blue corn extracts, compounds in the fat fraction are responsible for the antidiabetic activity (Smorowska, A. J., Å»oÅ‚nierczyk, A. K., Nawirska-OlszaÅ„ska, A., SowiÅ„ski, J., & Szumny, A. (2021). Nutritional properties and in vitro antidiabetic activities of blue and yellow corn extracts: A comparative study. Journal of Food Quality, 2021, 1-10.). In the research for the manuscript entitled Sensory and biological activity of medlar (Mespilus germanica) and quince 'Nivalis' (Chaenomeles speciosa): a comperative study, we did not study the antidiabetic properties of the oil fraction. I think that in future studies this aspect can be checked.

  • How much fat is in the skin and seeds? Why were these analysed?

In general, fruits and vegetables are low in fat (except for exceptions like avocados, nuts). As a rule, the largest amount of fat (with exceptions) is located in the seeds and peel of fruits and vegetables. The content of individual extracts in 2 kilograms of fruit is given in Figure 2. Masses of research material and obtained extracts (biomass, dry weight, amount of polyphenols, amount of fat). Based on this, it can be calculated that, for example, there are 58 milligrams of fat per kilogram of fruit biomass in quince peel.

Results and discussion

  • Is there any correlation between the linoleic acid and antidiabetic activity of the medler fruit?

There are many reports in the scientific literature regarding the anti-diabetic activity of linoleic acid. In the study conducted, we did not analyze this aspect. I think it is worth addressing this topic in future studies.

  • Line 362-364 if the skin is dangerous when raw why are you analysing it raw? Wasn't it supposed to be processed first for analysis? Stored fruits are still raw.

The analyzed medlar extracts with potential health-promoting properties were made on raw, lyophilized fruit parts. Treatment, such as heat, could cause the loss of biologically active compounds. Adding fruit pomace to tea can increase its nutritional value and introduce additional flavors and aromas. Medlar peels and seeds, are hard to digest, but extracts can find use in the food industry.

Conclusion

  • Is this study just to profile the fatty acids and sterols in medlar because the antioxidant activity and antidiabetic activity are well known (line 52-60)? Then the introduction should not be about antioxidant activity due to polyphenols but should justify the analysis of sterols and fatty acids and the conclusion should be on the purpose of this study.

In the cited publications (lines 52-60), whole fruit extracts were analyzed. In our manuscript, we separately extracted different parts of the fruit (peel, pulp and seeds) to determine the usefulness of the pomace, which is often a sluggish material.

Round 2

Reviewer 2 Report

The authors have revised to include some literature as requested.